# Adversarial Attack on Attackers: Post-Process to Mitigate Black-Box Score-Based Query Attacks

**Sizhe Chen**[1], **Zhehao Huang**[1], **Qinghua Tao**[2], **Yingwen Wu**[1], **Cihang Xie**[3], **Xiaolin Huang**[1]

[1]Department of Automation, Shanghai Jiao Tong University
[2]ESAT-STADIUS, KU Leuven
[3]Computer Science and Engineering, University of California, Santa Cruz

## Abstract

The score-based query attacks (SQAs) pose practical threats to deep neural networks by crafting adversarial perturbations within dozens of queries, only using the model's output scores. Nonetheless, we note that if the loss trend of the outputs is slightly perturbed, SQAs could be easily misled and thereby become much less effective. Following this idea, we propose a novel defense, namely Adversarial Attack on Attackers (AAA), to confound SQAs towards incorrect attack directions by slightly modifying the output logits. In this way, (1) SQAs are prevented regardless of the model's worst-case robustness; (2) the original model predictions are hardly changed, *i.e.*, no degradation on clean accuracy; (3) the calibration of confidence scores can be improved simultaneously. Extensive experiments are provided to verify the above advantages. For example, by setting $\ell_\infty = 8/255$ on CIFAR-10, our proposed AAA helps WideResNet-28 secure $80.59\%$ accuracy under Square attack (2500 queries), while the best prior defense (*i.e.*, adversarial training) only attains $67.44\%$. Since AAA attacks SQA's general greedy strategy, such advantages of AAA over 8 defenses can be consistently observed on 8 CIFAR-10/ImageNet models under 6 SQAs, using different attack targets, bounds, norms, losses, and strategies. Moreover, AAA calibrates better without hurting the accuracy.

## 1 Introduction

Deep Neural Networks (DNNs) are vulnerable to adversarial examples (AEs), where human-imperceptible perturbations added to clean samples can fool DNNs to give wrong predictions [1, 2]. Recently, such a threat is made practically feasible by the black-box score-based query attacks (SQAs) [3, 4, 5], as they only require the same information as users to craft efficient AEs. Users, for better judgments, need the model's prediction confidence indicated by DNNs' output scores, which is the only knowledge needed by SQAs to perform attacks. In contrast, white-box attacks [6, 7, 8] or transfer-based attacks [9, 10] require the gradients or training data of DNNs. Moreover, it has been shown that SQAs can achieve a non-trivial attack success rate by a reasonable number of queries, *e.g.*, dozens, compared to thousands of queries for decision-based query attacks [11, 12, 13]. Thus, such feasibility and effectiveness of SQAs are attracting increasing attention from defenders [14].

Defending against SQAs is a different goal compared to improving the *worst-case* robustness as commonly studied [15, 16, 17]. Because in *real-world scenarios* which SQAs are designed for, DNNs are treated as black boxes, in which the only interaction between models and users/attackers is the model's output scores. Thus, altering the scores is all defenders could do here, either in direct or indirect ways. Most existing defenses indirectly change outputs by optimizing the model [8, 18, 19] or pre-processing the inputs [14, 20], which, however, severely affect models' normal performance for different reasons. Training a robust model, *e.g.*, by adversarial training [8, 21, 22], diverts the model's attention to learning AEs, yielding the so-called accuracy-robustness trade-off

36th Conference on Neural Information Processing Systems (NeurIPS 2022).

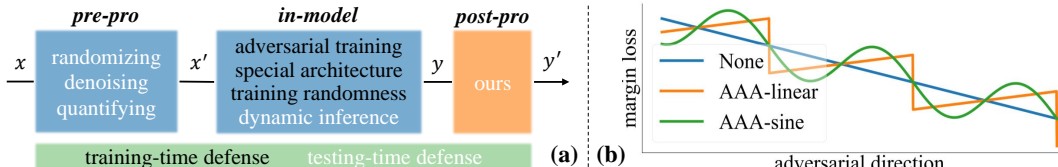

Figure 1: Compared to existing defenses on inputs or models, AAA post-processes to avoid SQAs as in (a). Our main idea is to show attackers the incorrect attack direction as in (b). Specifically, if we perturb the original undefended blue loss curve to the orange or green one, then attackers trying to decrease the loss would be mostly cheated away from their destiny, *i.e.*, the adversarial direction.

[23]. Randomizing [14, 24, 25, 26, 27] and blurring [20, 28, 29, 30] inputs reduce the signal-noise ratio, inevitably hurdling accurate decision. Dynamic inference [19, 31] is time-consuming due to the test-time optimization on model. Thus, it is imperative to develop a user-friendly defense against SQAs. In this paper, we hereby consider a post-processing defense as demonstrated in Fig. 1 (a), which naturally enjoys the benefits as follows: (1) model's decision is hardly affected since good predictions have already been obtained; (2) model calibration can be simultaneously improved via post-processing [32, 33] to output accurate prediction confidence; (3) it can be flexibly used as a plug-in module for pre-trained models with negligible test-time computation overhead. Despite these merits, it remains unexplored, to the best of our knowledge, that by post-processing,

*How to serve users while avoiding SQA attackers when they access the same output information?*

Since SQAs are black-box attacks that find the adversarial direction to update AEs by observing the loss change indicated by DNN's output scores, we can perturb such scores directly to fool attackers into incorrect attack tracks. Following this idea, we propose the adversarial attack on attackers (AAA), which manipulates the loss trend so that attackers, trying to greedily update AEs following the original trend, are led to an incorrect path. Specifically, AAA directly optimizes DNN's logits to control the output loss to approximate certain designed curves so that attackers, seeing our confounding outputs, would be attacked to lose their direction. In Fig. 1 (b), we give two representative designs of the loss curve to achieve the above goal. One is to increase loss mostly and decrease it dramatically and shortly along the correct attack direction, and another is to oscillate the curve around the original blue line. Now that a post-processing module is adopted here, we could simultaneously use it to calibrate the model as commonly proposed [32, 34]. Thus, a simple plug-in post-module is all you need to both fool attackers and offer users accurate confidence.

By post-processing, AAA not only lowers the calibration error without hurting accuracy in all cases, but also efficiently prevents SQAs, *e.g.*, helping a WideResNet-28 [35] on CIFAR-10 [36] secure $80.59\%$ accuracy under Square attack [5] (2500 queries), while the best prior defense (*i.e.*, adversarial training [37]) only attains $67.44\%$. Even if attackers try to guess defender's strategy (which requires heavy queries) and develop adaptive strategies, AAA could also confound both non-adaptive and adaptive attacks by the sine design. Because AAA attacks the general greedy update of SQAs, such advantages of AAA over 8 baseline defenses can be consistently observed on 8 CIFAR-10/ImageNet models under 6 SQAs, using different attack targets, bounds, norms, losses, and strategies, verifying AAA a user-friendly, effective, and generalizable defense. Our contributions are three-fold.

- We analyze current defenses from the view of SQA defenders and point out that a post-processing module forms not only effective but also user-friendly and plug-in defenses.
- We design a novel adversarial attack on attackers (AAA) defense that fools SQA attackers to incorrect attack directions by slightly perturbing the DNN output scores.
- We conduct comprehensive experiments showing that AAA outperforms the other 8 defenses in the accuracy, calibration, and protection from all tested 6 SQAs under various settings.

## 2  Related work

**Query attacks**  are black-box attacks that only require the model's output information. Query attacks can be divided into score-based query attacks (SQAs) [3, 4, 5, 38, 39, 40, 41] and decision-based query attacks (DQAs) [11, 12, 13, 42]. SQAs greedily update AEs from original samples by

observing the loss change indicated by DNN's output scores, *i.e.*, logits or probabilities. Early SQAs try to estimate DNN's gradients by additional queries around the sample [39, 43]. Recently, it is validated query-efficient to perform fast pure random-search SQAs [4, 5]. It has also been proposed to use pre-trained surrogate models in SQAs [40, 44], which, however, demands unfeasible access to DNN's training sample as in transfer-based attacks [9, 10]. Thus, we do not focus on defending such SQAs as also in [14]. Besides SQAs, DQAs rely only on DNN's decisions, *e.g.*, the top-1 predictions, to generate AEs. Since DQAs could not perform the greedy update, they start crafting the AE from a different sample and keep DNN's prediction wrong during the attack. Currently, DQAs need thousands of queries to reach a non-trivial attack success rate [45] compared to dozens of times for SQAs [5] as compared in Appendix A, limiting their threats. Additionally, SQAs are mostly applicable and attackers do not have to resort to DQAs because, in the real world, DNNs must not only be accurate but also indicate when they tend to be incorrect [32] by outputting confidence scores.

**Adversarial defense** mainly has two different goals. The first goal is to improve DNN's worst-case robustness [17], which demands DNNs to have no AE around clean samples bounded by a norm ball. Such robustness is originally evaluated by gradient-based white-box attacks [6, 7], but later, gradient obfuscation phenomenon is discovered [15], motivating evaluations to incorporate random noise and black-box attacks [16] in adaptive methods [16, 46, 47]. In this assessment, adversarial training (AT) [8, 21] is validated as the most effective defense as also against poisoning attacks [48, 49]. Other defenses, *e.g.*, using more or augmented data [50, 51], designing special architecture [18, 52, 53], and inducing randomness in training [54, 55], all need collaborations with AT to achieve good performance. Another defense goal is to mitigate real-case adversarial threats, and the defense performance is evaluated by the feasible and query-efficient SQAs. A more robust model in the worst cases certainly protects itself in real cases. Besides, it is also possible to defend by dynamic inference [19, 31], and randomizing [14, 24, 25, 26], denoising [20, 28] or quantifying [29, 30] inputs. However, all aforementioned defenses on inputs or models exert a non-negligible impact on accuracy, calibration, or inference speed due to the focus on learning AEs in AT, the reduction of signal-noise ratio in pre-processing, or the costly test-time optimization in dynamic inference as compared meticulously in Section 3.

**Model calibration** performance is a higher demand for a good model besides high accuracy. Besides correct predictions, model calibration additionally requires DNNs to produce accurate confidence in their predictions [56]. For instance, exactly $80\%$ of the samples predicted with $80\%$ confidence should be correctly predicted. In this point of view, the expected calibration error (ECE) [33] is widely used to quantify the error between accuracy and confidence. Various methods have been proposed to improve DNN's calibration in training [34, 57, 58, 59] or by a post-processing module after training [32, 33, 60]. Among them, temperature scaling [33] is a simple but effective method [32], which divides all output logits by a single scalar tuned by a validation set so that the ECE in testing samples would be significantly reduced. Since division is a simple post-processing operation, calibration could be simultaneously achieved by AAA, the first post-processing defense. There have been methods to avoid attacks by calibration [59] or calibrate by attack [34], but the simultaneous improvement of defense and calibration has not been reported to the best of our knowledge.

## 3 Preliminaries and motivation

Before presenting the proposed method, we first introduce the key ideas of SQAs and analyze existing defenses. For a sample $x$ labelled by $y$, an SQA on a DNN $f$ generates an AE $x'$ by minimizing the margin between logits [4, 5, 41, 61] as

$$\mathcal{L}(f(\boldsymbol{x}), y) = f_y(\boldsymbol{x}) - \max_{k \neq y} f_k(\boldsymbol{x}), \tag{1}$$

namely the margin loss [7]. For defenders without the label $y$, it is possible to calculate the unsupervised margin loss based only on the logits vector $\boldsymbol{z} = f(\boldsymbol{x})$ as

$$\mathcal{L}_{\mathrm{u}}(\boldsymbol{z}) \triangleq \mathcal{L}\left(f(\boldsymbol{x}), \hat{y}\right), \tag{2}$$

by assuming the model prediction $\hat{y}$ to be correct, since handling originally misclassified samples falls beyond the scope of defense [17]. For attackers with label $y$, it can be known that the attack succeeds if $\mathcal{L}(f(\boldsymbol{x}), y) < 0$. To quickly realize this, SQAs only update AEs if a query $\boldsymbol{x}_{\mathrm{q}}$ has a lower

margin loss compared to the current best query $\boldsymbol{x}_k$, *i.e.*,

$$\boldsymbol{x}_{k+1} = \begin{cases} \boldsymbol{x}_{\mathrm{q}}, & \mathcal{L}(f(\boldsymbol{x}_{\mathrm{q}}), y) < \mathcal{L}(f(\boldsymbol{x}_k), y), \\ \boldsymbol{x}_k, & \mathcal{L}(f(\boldsymbol{x}_{\mathrm{q}}), y) \geq \mathcal{L}(f(\boldsymbol{x}_k), y). \end{cases} \tag{3}$$

Existing SQAs [4, 5, 41] are quite effective, capable of halving the accuracy within 100 queries on CIFAR-10 when $\ell_\infty = 8/255$, posing significant and practical threats. Aware of this, the first defense specifically against SQAs has been recently proposed [14] to pre-process inputs by random noise. Besides, other existing defenses are also useful to avoid SQAs, among which the most representative ones are adversarial training (AT) [8, 21], dynamic inference [19, 31, 62], and pre-processing [14, 29]. These defenses work in different mechanisms as illustrated in Fig. 1 (a) and Section 2. Here in Table 1, we present a comparison of their key characteristics related to the defense performance.

Table 1: Expectation and effects of current defenses (unpreferable effects are marked in red)

|  | expectation | AT [8, 21] | pre-pro [14, 29] | dyn-inf [19, 31] | AAA (ours) |
|---|---|---|---|---|---|
| accuracy | = | ↓↓ | ↓ | = | = |
| calibration | ↑ | / | ↓ | ↓ | ↑ |
| testing cost | = | = | = | ↑↑↑ | = |
| training cost | = | ↑↑↑ | = | = | = |
| acc under SQA | ↑ | ↑↑ | ↑ | ↑ | ↑↑↑ |

Real-case defenders are expected to serve users and meanwhile avoid SQA attackers. The former demands good accuracy, calibration, and inference speed. While the latter requires good protection, and preferably, no additional training (denoted as "="). As a training-time defense on model, AT exerts a significant impact on accuracy [23], *e.g.*, >10% accuracy drop on ImageNet [52], with several-fold training computation and an indefinite impact (denoted as "/") on calibration. Pre-processing is a test-time defense on input, which avoids SQAs by reducing the input's signal-noise ratio but inevitably hurts normal performance. Recently, the dynamic inference is proposed as a test-time defense by optimizing the model, which shows no accuracy drop but dramatically increases the computation, *e.g.*, DENT [19] consumes >700% test-time calculation. Given the above discussion, it is imperative to develop a user-friendly and efficient method to effectively defend against SQAs.

## 4 Adversarial attack on attackers

Defending by post-processing naturally carries the advantages of good accuracy and calibration [32], no additional training and negligible test-time computation. Although it does not improve the commonly-studied worst-case robustness [16], its potential to defeat SQAs in real cases has not been explored yet to the best of our knowledge. By investigating current defenses from the perspective of SQA defenders in Sec. 3, it is interesting to find that in the real world, no matter whether defenders alter inputs, models, or outputs, what users/attackers get are just the changes in outputs, *i.e.*, the black-box setting. Thus, it is reasonable and preferable for SQA defenders to directly manipulate DNN's output scores, which has already become a standard practice in model calibration [32].

But how? Now that attackers conduct the adversarial attack on the model, why cannot we defenders actively perform an adversarial attack on attackers? If attackers search for the adversarial direction by scores, we can manipulate such scores to cheat them into an incorrect attack path. Following this idea, we develop the adversarial attack on attackers (AAA) to control what the SQA attackers base their actions on, *i.e.*, the loss (1) indicated by the DNN's output scores. Under such direct misleading, SQA attackers, following their original policy, would be guided to wherever we want them to go due to their greediness, seeing (3). But such change on outputs should be slight to ensure it is just attackers (observing loss trend) rather than users (observing outputs) that are misled. Therefore, the loss trend can not be reversed completely, but only be reversed locally in periodically-set intervals.

In each local interval, there are lots of designs to manipulate the loss trend. For example, we can let the loss along the adversarial direction (almost) always increase and only dramatically decrease between intervals, seeing the orange line of Fig. 1 (b). It is also possible to smooth the cross-interval loss change by making it oscillate like the green line in Fig. 1 (b), so that attackers could not easily

guess defender's strategy by observing the loss drop. Either design cheats attackers by slight output modifications so that users get accurate scores. This is because we manually divide loss values into small intervals and handle them separately, which is realized by designing the misleading loss curve based on periodic *loss attractors* as

$$l_{\text{atr}} = (\text{floor}\,(l_{\text{org}}/\tau) + 1/2) \times \tau, \tag{4}$$

where $l_{\text{org}} = \mathcal{L}_{\text{u}}(z_{\text{org}})$ is the original loss for the unmodified logits $z_{\text{org}}$. $\tau$ is the period of attractor intervals, and "floor" denotes rounding down decimals to integers. Eq. (4) sets loss values from $[l_{\text{atr}} - \tau/2, l_{\text{atr}} + \tau/2]$ as an interval. For example, if $\tau = 6$, then according to Eq. (4), the closest loss attractor of logits with $l_{\text{org}} = 0 \to 6$ is $l_{\text{atr}} = 3$. Thus, the $1/2$ term is necessary to avoid setting $l_{\text{atr}} = 0$, *i.e.*, the decision boundary, which may flip model's decision.

Periodically-set loss attractors divide logits with different $l_{\text{org}}$ into intervals, so that we could manipulate the loss trend in each interval to form our designed loss curve that cheats attackers. The target loss values $l_{\text{trg}}$ in the two misleading curves $l_{\text{trg\_lnr}}, l_{\text{trg\_sin}}$ in Fig. 1 (b) can be expressed as

$$
\begin{aligned}
l_{\text{trg\_lnr}} &= l_{\text{atr}} - \alpha \times (l_{\text{org}} - l_{\text{atr}}) \\
l_{\text{trg\_sin}} &= l_{\text{org}} - \alpha \times \tau \sin(\pi(1 - 2(l_{\text{org}} - l_{\text{atr}})/\tau)).
\end{aligned}
\tag{5}
$$

In the former linear design, when $l_{\text{org}}$ decreases from $l_{\text{atr}} + \tau/2$ to $l_{\text{atr}} - \tau/2$ as the sample approximates the decision boundary, we output loss that increases from $l_{\text{atr}} - \alpha \times \tau/2$ to $l_{\text{atr}} + \alpha \times \tau/2$ to fool attackers that this is an incorrect attack direction. In contrast, AAA-sine outputs an increasing loss when $l_{\text{org}}$ decreases from around $l_{\text{atr}} + \tau/4$ to $l_{\text{atr}} - \tau/4$. Although we do not always reverse the loss trend here, it is sufficient to resist SQAs. Plus, AAA-sine enjoys an additional advantage to fool attackers that try to guess our defense strategy due to the smooth loss transition across intervals.

Although the periodic design has already altered output confidence very slightly, we could also step further to improve the precision of confidence because post-processing logits is a standard practice in model calibration [33, 60]. Thus, simultaneous defense and calibration is obtainable in a single AAA module by controlling the loss while encouraging the output confidence $\sigma(z) = \max(\text{softmax}(z))$, *i.e.*, the maximum probability after softmax, to approach the calibrated one $p_{\text{trg}}$ as

$$\min_{z} \quad \|\,\mathcal{L}_{\text{u}}(z) - l_{\text{trg}}\,\|_1 + \beta \cdot \|\,\sigma(z) - p_{\text{trg}}\,\|_1, \tag{6}$$

which is a straightforward design to fulfill two purposes. The first term encourages the perturbed logits $z$ to have a loss $L_{\text{u}}(z)$ close to the target value $l_{\text{trg}}$, forming the misleading loss curve for SQAs. The second item motivates the output confidence $\sigma(z)$ close to the calibrated one $p_{\text{trg}}$ so that users get accurate confidence scores. $\beta$ balances between the above two goals. Despite its simplicity, Eq. (6) has to be solved by optimization because the exponential operation in softmax makes Eq. (6) a transcendental equation without closed-form solutions. Luckily, optimizing low-dimensional logits is not costly [33, 60]. By optimizing Eq. (6) for $\kappa$ iterations, DNN's output is both accurate and misleading, achieving two seemingly contradictory goals simultaneously.

The calibrated confidence $p_{\text{trg}}$ is obtainable by various model calibration methods [33, 60]. Among them, temperature scaling [33] is simple but effective [32], which divides all logits by a scalar $T$ as $p_{\text{trg}} = \sigma(z_{\text{org}}/T)$. The temperature $T$ is tuned by a validation set to minimize the calibration error, and here, such tuning is conducted when the optimization (6) is also performed for $\kappa$ iterations so that we can find the temperature that suits AAA best. After that, $T$ is fixed for inference in AAA.

We summarize our algorithm in Alg. 1. AAA first calculates the original loss $l_{\text{org}}$, according to which sets the target loss $l_{\text{trg}}$ that forms the misleading loss curve. Then AAA optimizes the logits to reach $l_{\text{trg}}$, and also the target confidence $p_{\text{trg}}$, which is obtained by dividing the original logits using a pre-tuned temperature $T$. The overall procedure is in test-time with only optimization on the logits, making AAA a computation-efficient,

---

**Algorithm 1** Adversarial Attack on Attackers

**Input:** the logits $z_{\text{org}}, T, \tau, \alpha, \beta, \kappa$.
**Output:** post-processed logits $z$
  1: get original loss $l_{\text{org}} = \mathcal{L}_{\text{u}}(z_{\text{org}})$ by (2)
  2: set target loss $l_{\text{trg}}$ by (5)
  3: set target confidence $p_{\text{trg}} = \sigma(z_{\text{org}}/T)$
  4: initialize $z = z_{\text{org}}$ and optimize it for (6)
  5: **return** $z$

---

plug-in, and model-agnostic method with good accuracy, calibration, and defense against SQAs.

# 5 Experiments

## 5.1 Setup

We evaluate AAA along with 8 defense baselines, including random noise defense (RND [34]), adversarial training (AT [37, 63, 52]), dynamic inference (DENT [19]), training randomness (PNI [54]), and ensemble (TRS [64]). Results of AT with extra data [23]) are in Appendix A. For AAA, we mostly experiment with the linear design, because the two designs of AAA share most good characteristics. AAA-sine could fool adaptive attacks, which is costly to design in real world as discussed in Sec 5.6. AAA-linear uses $\alpha = 1, \tau = 6, \beta = 5, \kappa = 100$, and an Adam optimizer [65] with learning rate $0.1, \beta_1 = 0.9, \beta_2 = 0.999$. AAA-sine uses $\alpha = 0.7$ and inherits other hyper-parameters in study of adaptive attacks. To calibrate simultaneously, we perform temperature scaling using 1K/5K samples for CIFAR-10 testing set / ImageNet validation set to and make them disjoint with the attack samples as much as possible. Other hyper-parameters are in Appendix E.

The defenses are assessed by 6 state-of-the-art SQAs, which are random search methods including Square [5], SignHunter [41], and SimBA [4], and gradient estimation methods including NES [61], and Bandits [3]. Square attack [5] selects a decreasing size of random square perturbations. SignHunter [41] estimates the sign of gradient and flips corresponding perturbations. SimBA [4] randomly samples perturbations given an orthonormal basis. NES [61] adopts the Natural Evolutionary Strategies for gradient estimation. Bandits [3] jointly uses the time and data-dependent gradient prior. Some SQAs use pre-trained models [40, 44], which, however, demands unfeasible access to DNN's training sample. To evaluate real-case defenses against training-based attacks, we alternatively consider a practical SQA QueryNet [66] based on model stealing. Results of DQAs and SQAs using other than margin loss are put in Appendix A. We mostly perform untargeted $\ell_\infty$ attacks, but we also test the targeted and $\ell_2$ attacks under different bounds to observe AAA's generalization.

We use 8 DNNs, which are mostly WideResNets [35] as in [17]. The pre-trained models of PNI [54] / TRS [64] are ResNet-20 [67], and we also test ResNeXt-101 [68] in ImageNet. Other studied DNNs come from RobustBench [17] and torchvision [69] as specified in Appendix E. AT models are tested using the same bound as in AT, if not otherwise stated. We use all 10K CIFAR-10 testing samples. For ImageNet, we randomly-select 1K validation samples from all 1K classes respectively (1 image in 1 class) to eliminate the class bias as in [66]. All images are rescaled to $[0, 1]$, and the ImageNet ones are resized to $224 \times 224$. Before feeding them to DNNs, we quantify images to 8-bit, imitating the real-case 8-bit image setting [66]. Experiments are performed on an NVIDIA Tesla A100 GPU (a GPU with 4GB+ memory works). Our code is available at `https://github.com/Sizhe-Chen/AAA`.

As defenders, we are concerned about DNN's remaining accuracy after it being attacked by SQAs for a certain query times. Thus, we report such *SQA adversarial accuracy* after 100 and 2500 queries, reflecting DNN's performance under mild and extreme SQAs. We include the average query times, another metric commonly used by attackers, in Appendix C. To measure the calibration, the expected calibration error (ECE) [33] is commonly used. ECE divides all $N$ testing samples into $M$ bins, and each bin contains samples with confidence ranging from the $m^{\text{th}}$ quantile to the $(m + 1)^{\text{th}}$ quantile of $[0, 1]$. Then ECE is calculated by the difference between accuracy and confidence as $\frac{1}{N} \sum_{m=1}^{M} \left| \sum_{i \in B_m} \mathbf{1}(\hat{y}_i = y_i) - \sum_{i \in B_m} \hat{p}_i^{\hat{y}_i} \right|$, where $\hat{y}_i$ is the predicted label of the $i^{\text{th}}$ sample in the $m^{\text{th}}$ bin $B_m$, and $\hat{p}_i^{\hat{y}_i}$ represents the probability confidence of this prediction.

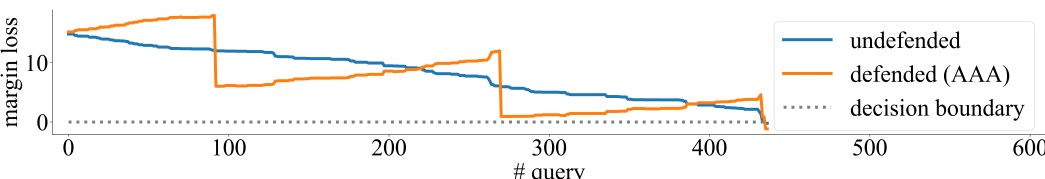

Figure 2: The AAA(linear)-defended and undefended margin loss value when attacking the undefended WideResNet-28 [35] in RobustBench [17] by Square attack [5] ($\ell_\infty = 8/255$) using the $9953^{\text{th}}$ CIFAR-10 test sample (other samples have similar trends as in Appendix B). AAA fools attackers precisely as shown by the tiny symmetric oscillations of two lines.

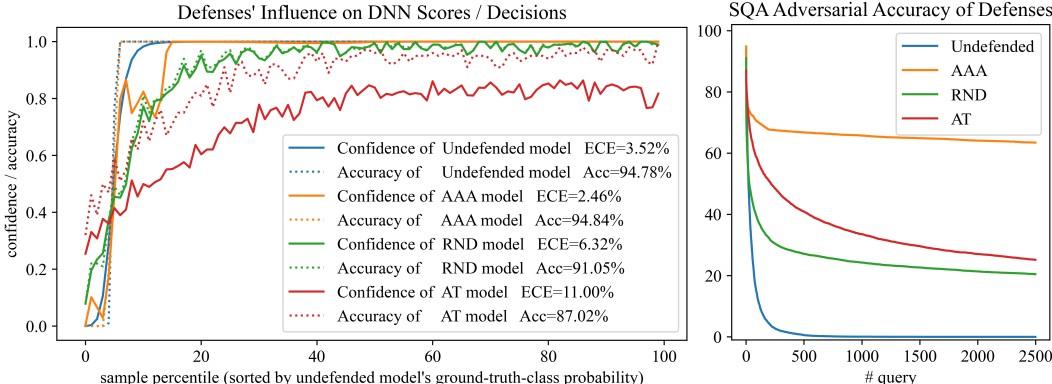

Figure 3: The left figure illustrates the change of output scores for different defenses. We first sort 10K testing samples in ascending order according to their ground-truth-class probability predicted by the undefended model (blue line), and then divide them into 100 bins, so that the left bins in the figure stand for low-confidence or misclassified samples, and vice versa. Then for samples in each bin, we plot the confidence (solid lines) and accuracy (dotted line) for AT [37], RND [34], and AAA-linear . Compared to AT and RND, AAA alters scores most slightly without influencing the accuracy. The right figure shows DNN's accuracy under Square attack [5] when $\ell_2 = 2.5$, indicating that AAA outperforms alternatives in defending SQAs by a large margin with also the highest clean accuracy.

## 5.2 Visual illustration of AAA

We first visually illustrate the mechanism and effects of AAA. AAA, besides fooling attackers, also calibrates the model, forming the loss curve as in Fig. 2. Slightly different from the ideal defense in Fig. 1 (b) is that the orange curve is lowered for calibration. However, calibration does not hurdle fooling attackers, *i.e.*, the orange line is mostly going opposite to the blue line in a precisely symmetric manner. Also, two lines cross the decision boundary at the same time, meaning that AAA does not change the decision (the negligible increase of AAA accuracy comes from randomness).

Besides the loss, it is also necessary to observe the output probabilities of AAA, which is displayed in Fig. 3 (left). AAA, indicated by the orange lines, changes the scores slightly without hurting accuracy (dotted lines) compared to RND [34] and AT [37]. By such small modifications, AAA not only improves the calibration but also prevents SQAs by misleading them into incorrect directions. As displayed by the right part of Fig. 3, AAA is super effective in avoiding SQAs, even if it is a query-efficient attack working at a large bound. Specifically, AAA preserves a standard trained DNN to have >60% accuracy after 2500 queries, doubling the performance of AT and tripling that for RND.

## 5.3 Numerical results of AAA

We report the main numerical results in Table 2, where RND and AAA-linear are directly implemented in the undefended model denoted as "None". The AT method for the three models are PSSiLU [37], vanilla AT [63], and feature denoising [52], respectively. According to Table 2, AAA-linear not only consistently reduces ECE by >12%, but also does not hurt clean accuracy. In contrast, the AT models lose >7% accuracy and RND also endures a drop in accuracy and calibration. Under SQAs, AAA-linear preserves a significantly higher accuracy than the undefended model, which is totally destroyed. The most threatening SQAs in two datasets more than halve the CIFAR-10 model's accuracy within 100 queries and degrade ImageNet ones to <15% by 2500 times. However, the AAA model remains >75% and >55% accuracy in extreme cases. AT and RND are useful in mitigating SQAs, but it is AAA-linear that tops the defense performance in almost all cases.

Besides AT and RND, diverse defenses have also been proposed. DENT [19] optimizes the model in test-time. PNI [54] injects noise during training. TRS [64] ensembles three models with low attack transferability. They are developed for gradient-based attacks, but also provide protection against SQAs. However, seeing Table 3, they are not comparable to AAA-linear in real cases regarding the accuracy, calibration, and defense performance. Here, we also test a strong SQA QueryNet, which uses three architecture-alterable models to steal the DNN. Due to its utilization of large-scale testing samples, QueryNet greatly hurts DNNs, but AAA is still the defense that protects the model best.

Table 2: The defense performance under attacks (#query = 100/2500)

| Model | Metric / Attack | None | AT [37, 63, 52] | RND [14] | AAA-linear |
|---|---|---|---|---|---|
| CIFAR-10 $\ell_\infty = \frac{8}{255}$ | ECE (%) | 3.52 | 11.00 | 6.32 | **2.46** |
| | Acc (%) | 94.78 | 87.02 | 91.05 | **94.84** |
| Wide-ResNet-28 [35] | Square [5] | 39.38 / 00.09 | 78.30 / 67.44 | 60.83 / 49.15 | **81.36 / 80.59** |
| | SignHunter [41] | 41.14 / 00.04 | 78.87 / 66.79 | 61.02 / 47.82 | **79.41 / 76.71** |
| | SimBA [4] | 53.04 / 03.95 | 84.21 / 75.85 | 76.39 / 64.34 | **88.86 / 83.36** |
| | NES [61] | 83.42 / 12.24 | 85.92 / 81.01 | 86.23 / 68.19 | **90.62 / 85.95** |
| | Bandit [3] | 69.86 / 41.03 | **83.62** / 76.25 | 70.44 / 41.65 | 80.86 / **78.36** |
| ImageNet $\ell_\infty = \frac{4}{255}$ | ECE (%) | 5.42 | 5.03 | 5.79 | **4.30** |
| | Acc (%) | 77.11 | 66.30 | 75.32 | **77.17** |
| Wide-ResNet-50 [35] | Square [5] | 52.27 / 09.25 | 59.20 / 51.11 | 58.67 / 50.54 | **63.13 / 62.51** |
| | SignHunter [41] | 53.05 / 13.88 | 59.47 / 56.22 | 59.36 / 52.98 | **62.35 / 56.80** |
| | SimBA [4] | 71.79 / 20.90 | 65.64 / 47.60 | 66.36 / 63.27 | **74.16 / 67.14** |
| | NES [61] | 77.11 / 64.93 | 66.30 / 64.38 | 71.33 / 66.05 | **77.12 / 67.06** |
| | Bandit [3] | 71.33 / 65.77 | 65.30 / 63.98 | 65.15 / 61.38 | **72.15 / 70.53** |
| ImageNet $\ell_\infty = \frac{4}{255}$ | ECE (%) | 8.37 | **5.74** | 8.93 | 7.38 |
| | Acc (%) | **78.21** | 63.94 | 76.75 | **78.21** |
| Res-NeXt-101 [68] | Square [5] | 54.51 / 11.73 | 57.99 / 51.47 | 58.56 / 48.20 | **66.32 / 65.77** |
| | SignHunter [41] | 54.12 / 13.30 | 59.14 / 56.49 | 58.02 / 52.50 | **62.72 / 59.60** |
| | SimBA [4] | 70.86 / 24.64 | 59.40 / 57.53 | 68.31 / 66.16 | **74.06 / 67.26** |
| | NES [61] | **78.21** / 66.24 | 63.94 / 62.51 | 73.37 / **69.30** | **78.21** / 68.51 |
| | Bandit [3] | 72.19 / 67.73 | 63.18 / 62.26 | 67.69 / 64.62 | **72.91 / 71.48** |

Table 3: Various defenses under strong attacks (#query = 100/2500, CIFAR-10, $\ell_\infty = 8/255$)

| Metric / Attack | None | DENT [19] | PNI [54] | TRS [64] | AAA-linear |
|---|---|---|---|---|---|
| ECE (%) | 3.52 | 5.20 | 3.09 | 3.64 | **2.46** |
| Acc (%) | 94.78 | 94.80 | 81.91 | 88.64 | **94.84** |
| Square [5] | 39.38 / 00.09 | 62.01 / 35.07 | 57.69 / 45.47 | 56.26 / 23.91 | **81.36 / 80.59** |
| QueryNet [41] | 13.50 / 00.03 | 39.35 / 20.75 | 44.06 / 34.69 | 16.43 / 08.49 | **50.01 / 49.63** |

## 5.4 Generalization of AAA

Aside from the untargeted $\ell_\infty$ attacks, we also conduct targeted attacks, $\ell_2$ attacks under different bounds to study the generalization of AAA. A targeted attack is successful only if an AE is mispredicted as a pre-set class, which, is randomly chosen from incorrect classes for each sample here. And $\ell_2$ attacks bound the perturbations by $\ell_2$ norm, which is reported to fool DNNs better [70]. Here we additionally validate AAA's plug-in advantage by combining it with AT. Note that what differs from AAA and most existing defenses [52, 53, 54] combined with AT is that AAA has already achieved excellent defense performance, so such a combination is feasible but not necessary. We choose Square attack [5] to perform the above evaluations because it is the most effective SQA as tested in Table 2. The results are presented in Table 4, where all models are exactly the same as in Table 2 (CIFAR-10) without tuning hyper-parameters of defense to fairly evaluate the generalization of different methods. The results of $\ell_2$ AT model and other $\ell_\infty$ AT models are put in Appendix A.

In the difficult targeted attack setting, the undefended model remains only $2.84\%$ accuracy after 2500 queries, which is approximately just the accuracy drop of the AAA model. For $\ell_2$ attacks, AAA-linear is still capable of mitigating threats without hurting users, and its superiority is more outstanding as the attack bound becomes larger. AT models, although robust, suffer from attacks under a large or different norm ball [59]. Thus, its defense effects decrease as SQA alters the setting, seeing the bottom line. Defended by AAA, however, this drawback would be greatly avoided. An $\ell_\infty$ AT model, even under $\ell_2 = 2.5$ attack after 2500 queries, is hardly influenced, *i.e.*, increasing query times after 100 queries hardly better the attack performance, discouraging SQA attackers.

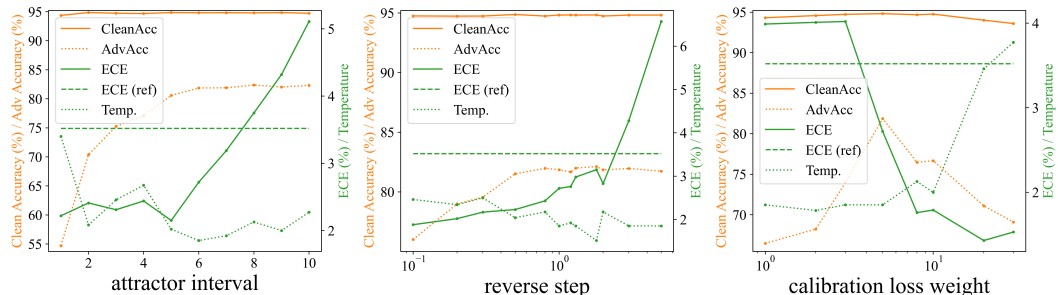

Figure 4: The influence of AAA hyper-parameters (the attractor interval $\tau$ in (4), the reverse step $\alpha$ in (5), and the calibration loss weight $\beta$ in (6)) to accuracy (left axis), ECE (right axis), SQA adversarial accuracy (left axis), and temperature (right axis). The dashed ECE (ref) means the undefended model.

Table 4: Generalization of AAA tested by Square attack [5] (#query = 100/2500, CIFAR-10)

| Metric / Attack | None | AAA-linear | AT [37] | AT-AAA-linear |
|---|---|---|---|---|
| ECE (%) | 3.52 | **2.46** | 11.00 | **10.56** |
| Acc (%) | 94.78 | **94.84** | 87.02 | **87.02** |
| untargeted $\ell_\infty = 8/255$ | 39.38 / 00.09 | **81.36 / 80.59** | 78.30 / 67.44 | **80.80 / 80.13** |
| targeted $\ell_\infty = 8/255$ | 75.59 / 02.84 | **92.05 / 91.62** | 85.75 / 82.72 | **86.22 / 86.13** |
| untargeted $\ell_2 = 0.5$ | 81.53 / 18.75 | **92.66 / 92.63** | 84.26 / 78.97 | **85.12 / 84.31** |
| untargeted $\ell_2 = 2.5$ | 12.77 / 00.01 | **70.35 / 63.46** | 57.88 / 25.19 | **74.03 / 73.72** |

## 5.5 Hyper-parameters of AAA

The reported amazing results in various settings are all obtained using fixed parameters that are heuristically selected, and it would be necessary to see how each of them affects the results. The attractor interval $\tau$ in (4) decides the period of the margin loss attractors, and a larger $\tau$ divides the losses into fewer intervals so that attackers are harder to jump out of one of them. The reverse step $\alpha$ in (5) controls the reverse degree, and if it increases, the modified loss curve would be steeper oppositely along the adversarial direction, emphasizing the defense. The calibration loss weight $\beta$ in (6) directly balances defense and calibration in optimization. Here we still consider the accuracy, ECE, and SQA adversarial accuracy (under 100 queries). Plus, we study an additional metric, the temperature $T$ of logits rescaling tuned with AAA-linear.

The results are shown in Fig. 4. The adversarial accuracy far exceeds the undefended model ($39.38\%$) and mostly surpasses AT ($67.44\%$). The clean accuracy is hardly impacted and the ECE is mostly below the undefended model (the green dashed line). Thus, AAA's good performance is insensitive to hyper-parameters, even if tuned in logarithmic scale. Regarding the trend, an intuitive conclusion is that a larger $\alpha$, a larger $\tau$, or a smaller $\beta$ that highlights defense more v.s. calibration would thereby increase the adversarial accuracy and ECE. Interestingly, the temperature mostly decreases as the defense is emphasized (the logits are divided by a smaller value), indicating that the model tends to output lower confidence to defend, consistent with the situation in AT [17].

We also study the influence of optimization times $\kappa$ in AAA to see the balance between the amount of computation and defense performance. AAA's runtime is estimated by inferring 10K CIFAR-10 samples in a WideResNet28 model on an NVIDIA GeForce RTX 2080Ti GPU. The results are shown in Table 5. As we see, optimizing low-dimensional logits is not costly, and good defense and calibration results are also obtainable by 60 to 80 iterations, which costs less time. Since optimizing logits is independent of model size, model owners could determine AAA runtime precisely.

Table 5: Influence of the optimization times in AAA (100-query Square attack on CIFAR-10)

| $\kappa$ | 0 | 20 | 40 | 60 | 80 | 100 |
|---|---|---|---|---|---|---|
| ECE | 3.52 | 2.87 | 2.81 | 2.66 | 2.53 | 2.53 |
| Adv-Acc | 39.38 | 79.29 | 80.92 | 81.37 | 81.28 | 81.36 |
| inference time per sample (ms) | 1.016 | 1.034 | 1.088 | 1.099 | 1.143 | 1.163 |

### 5.6 Adaptive attacks of AAA

Designing adaptive attacks, though possible, is costly and easy to bypass in real-world scenarios. Because here, attackers and defenders are in a double-blind relationship, i.e., attackers do not know the model, including the defense strategy. Thus, the discovery process of defense strategies for developing adaptive attacks would require additional queries. After that, attackers also have to devote considerable manual efforts to creatively deduce what defenders actually do. In this regard, AAA-linear has already imposed a great hurdle to adaptive attackers.

But still, it is interesting to see that after unveiling defender's strategy by great effort, how adaptive attackers could break the AAA model. There are two straightforward adaptive attacks against AAA: going bidirectional or opposite to the original attack direction, which are both based on the exceptional loss change that attackers observe in AAA-linear to decide the update direction. However, adaptive attackers are also easy to fool because the dramatic drop of loss across intervals could be smoothed, e.g. by using a sine function to design the target loss as the second equation in (5). In this way, neither direction of search is likely to figure out the defense strategy, seeing results below.

The bidirectional / opposite search weakens AAA-linear from 81.36% to 62.91% / 57.31%, but AAA-sine that ascends and descends the loss along the attack direction misleads both non-adaptive and adaptive SQAs. By a sine $l_{\mathrm{trg}}$, the defended adversarial accuracy for the non-adaptive attack

Table 6: AAA under adaptive attacks (100 queries)

| Defense | None | AAA-linear | AAA-sine |
|---------|------|------------|----------|
| Square | 39.38 | 81.36 | 78.34 |
| bi-Square | 57.09 | 62.91 | 76.69 |
| op-Square | 94.78 | 57.31 | 76.41 |

is kept at 78.34% while that under adaptive attacks is improved to 76.69% / 76.41%. Thus, defenders could easily mitigate even adaptive SQAs following our idea to fool attackers.

## 6 Conclusion, impacts and limitations

We develop a novel defense against score-based query attacks (SQAs). Our main idea is to actively attack the attackers (AAA), misleading them into incorrect attack directions. AAA achieves that by post-processing DNN's logits while enforcing the new output confidence to be calibrated, making AAA a deterministic plug-in test-time defense with improvements in calibration and accuracy by costing negligible computation overhead. Compared to alternative defenses, AAA is effective in mitigating SQAs according to our study on 8 defenses, 6 SQAs, and 8 DNNs under various settings.

As a defense in real-world applications, AAA greatly mitigates the adversarial threat without requiring huge computational burden. For example, in autonomous driving or supervision systems, the post-processing defense module could be directly implemented in pre-trained models. Despite the low cost, the benefits of adopting AAA are profound. In most cases, users would be provided with a more accurate confidence score so that they know better when the model tends to fail. In adversarial cases, SQAs, the most threatening attack in real cases, would be effectively prevented.

AAA is developed to especially prevent SQAs. Thus, defending other types of attacks is beyond our scope. For example, AAA does not improve the worst-case robustness evaluated in white-box settings [16, 47] where attackers have complete knowledge of the model (the AutoAttack [16] robust accuracy would be increased in an undesirable manner by AAA). Also, AAA is not applicable to avoiding transfer-based attacks and decision-based query attacks, which are either unfeasible or inefficient in the real world, because AAA induces only a negligible impact on the decision boundary.

## Acknowledgments and Disclosure of Funding

This work was partially supported by National Natural Science Foundation of China (61977046), Research Program of Shanghai Municipal Science and Technology Committee (22511105600), and Shanghai Municipal Science and Technology Major Project (2021SHZDZX0102). Cihang Xie is supported by a gift from Open Philanthropy. The authors are grateful to the anonymous reviewers for their insightful comments.

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
