# A More numerical results

**Decision-based attacks.** In Table 7, we present results concerning a SOTA decision-based attack, RamBo [45] (denoted as "RB"). As shown in the first two columns, compared to Square [5] (denoted as SQ), RamBo could hardly impact the undefended DNN even after 2500 queries under a large $\ell_2 = 1.5$ bound. Thus, it is reasonable for us to target at mitigating black-box SQAs in real cases.

**$\ell_2$ attacks.** Seeing the right four columns of Table 7, one could observe that when the attack bound increases, both $\ell_\infty$ and $\ell_2$ AT models are impacted much more significantly. Moreover, AAA's superiority is enhanced as the attack becomes stronger.

Table 7: The adversarial accuracy under RamBo (RB) [45] and Square (SQ) [5] (#query = 2500)

| Bound | None-RB | None-SQ | $\ell_\infty$ AT [37]-SQ | $\ell_2$ AT [23]-SQ | RND-SQ | AAA-SQ |
|---|---|---|---|---|---|---|
| $\ell_2 = 0.5$ | 94.78 | 18.75 | 78.97 | 85.64 | 66.18 | **92.63** |
| $\ell_2 = 1.0$ | 94.74 | 02.22 | 66.79 | 76.34 | 59.09 | **90.01** |
| $\ell_2 = 1.5$ | 94.10 | 00.31 | 50.97 | 61.64 | 46.52 | **82.38** |
| $\ell_2 = 2.0$ | 69.50 | 00.02 | 36.24 | 45.22 | 33.35 | **72.99** |
| $\ell_2 = 2.5$ | 10.38 | 00.01 | 25.19 | 28.56 | 20.47 | **63.46** |

**Attack using different losses.** Although attackers generally greedily update based on the margin (of logits) loss [5, 41, 61], it is possible for them to choose other loss options such as minimizing the probability margin and maximizing the cross-entropy loss. The results in Table. 8 show that despite the choice of AAA to reverse the margin loss, it could prevent attacks using other loss types.

**The plug-in advantage of AAA.** AAA, as a plug-in post-processing defense, is embeddable into any defense that increases the model's robustness. As shown in Table 8, AAA dramatically decreases the ECE of AT models without impacting the accuracy. Moreover, the already good defense performance of AT models is further boosted by AAA.

Table 8: The defense performance under Square attack [5] (#query = 100/2500)

| Metric / Loss | CIFAR-10 ($\ell_\infty = 8/255$) WideResNet34 [35] | | ImageNet ($\ell_\infty = 4/255$) WideResNet50 [35] | |
|---|---|---|---|---|
| | AT [23] | AT-AAA-linear | AT [63] | AT-AAA-linear |
| ECE | 18.96 | **5.93** | 5.03 | **2.64** |
| Acc | **91.47** | **91.47** | **66.30** | **66.30** |
| logits-margin | 83.22 / 69.67 | **84.68 / 82.92** | 59.20 / 51.12 | **60.83 / 59.73** |
| probability-margin | 82.90 / 69.38 | **84.41 / 82.67** | 59.21 / 50.59 | **60.33 / 57.88** |
| cross-entropy | 83.93 / 71.17 | **84.55 / 82.57** | 60.13 / 52.84 | **60.53 / 58.01** |

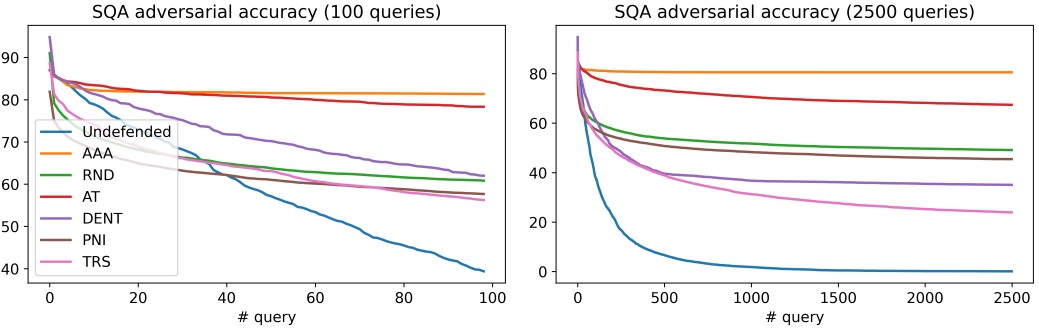

Figure 5: The adversarial accuracy under Square attack [5] when $\ell_\infty = 8/255$, indicating that AAA outperforms alternatives in defending SQAs by a large margin with also the highest clean accuracy.

# B   More visual illustrations

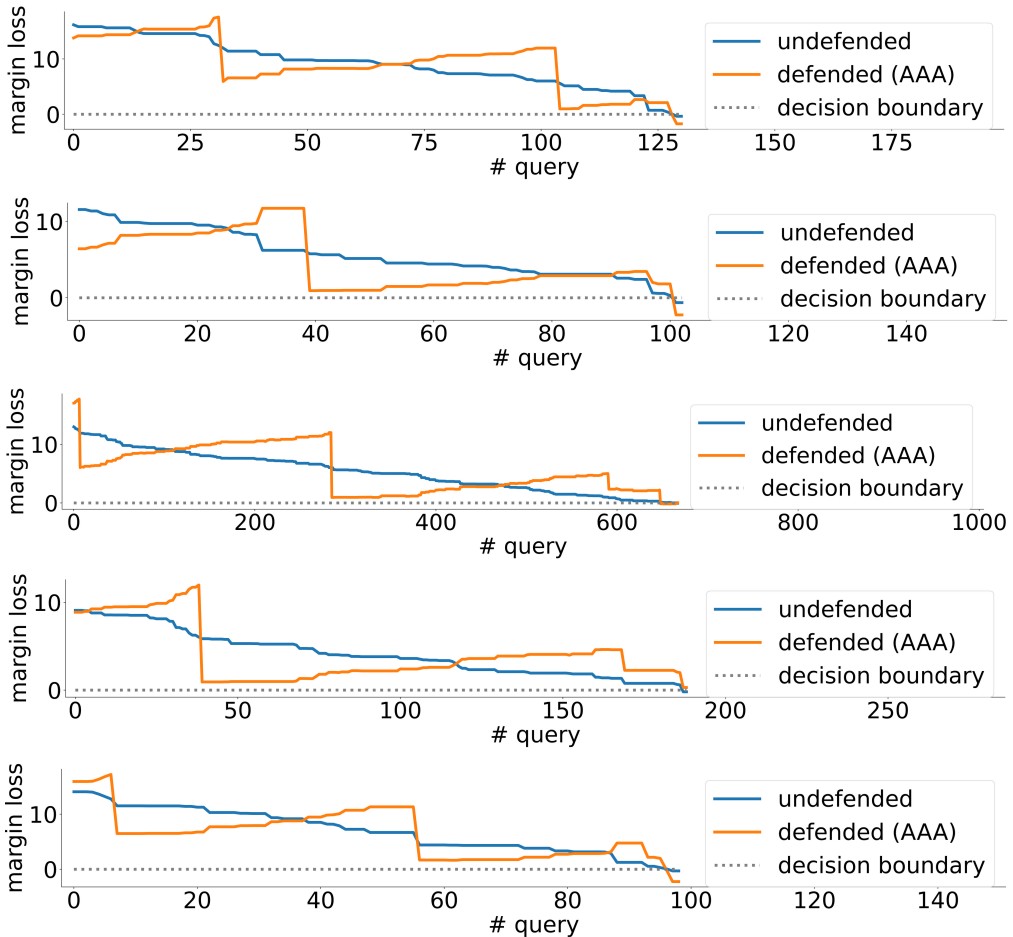

Figure 6: The AAA-defended and undefended margin loss value when attacking the undefended WideResNet-28 [35] in RobustBench [17] by Square attack [5] ($\ell_\infty = 8/255$) using the $1^{rd}, 2^{nd}, 5^{th}, 7^{th}, 8^{th}$ CIFAR-10 test sample.

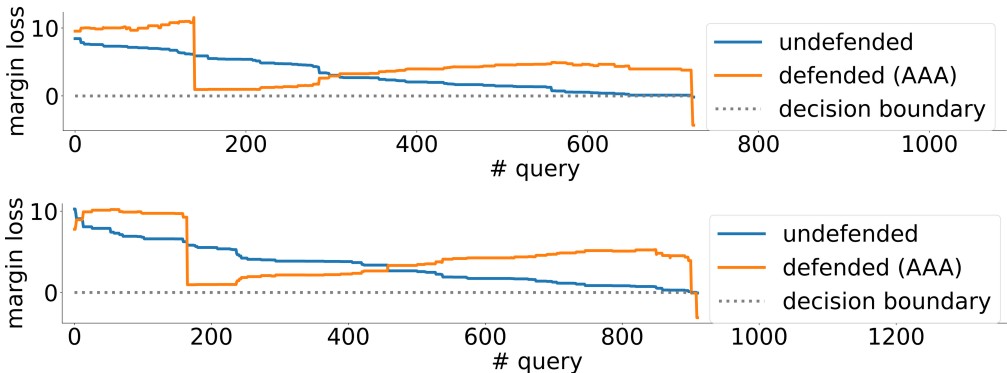

Figure 7: The AAA-defended and undefended margin loss value when attacking the undefended WideResNet-50 [35] in RobustBench [17] by Square attack [5] ($\ell_\infty = 4/255$) using the $2^{nd}, 7^{th}$ ImageNet test sample.

## C   Error analysis

**Results of multiple runs.**   We run the main experiments in Table 2 for 5 times using different random seeds for Square attack, and report the results in Table 9. Since AAA is a deterministic method without randomness, its defense performance is very constant.

**Average query times.**   Attackers generally use the average query times (AQ) to measure attacks, which is reported in Table 9. Here we record the AQ of all query samples to reflect the real attack cost. AAA hurdles the attack very much, seeing the large AQ and adversarial accuracy.

**Computation.**   We report the FLOPs of each model in Table 2. Since the only calculation of AAA is to post-process logits, the computational overhead is negligible (<0.01 GFLOPs). The total amount of required calculation could be obtained by multiplying FLOPs with AQ using 10K samples.

Table 9: The defense performance under Square attack [5]

| Dataset | CIFAR-10 | ImageNet | ImageNet |
|---|---|---|---|
| Model | WideResNet-28 [35] | WideResNet-50 [35] | ResNeXt-101 [68] |
| Acc (%) | 94.84 | 77.17 | 78.32 |
| ECE (%) | 2.46 | 4.30 | 7.38 |
| FLOPs (G) | 5.24 | 11.43 | 16.48 |
| Bound ($\ell_\infty$) | 8/255 | 4/255 | 4/255 |
| AdvAcc-100 | $81.77 \pm 0.24$ | $63.77 \pm 0.40$ | $66.63 \pm 0.48$ |
| AdvAcc-2500 | $81.00 \pm 0.25$ | $62.89 \pm 0.27$ | $66.13 \pm 0.44$ |
| AQ-100 | $87.00 \pm 0.26$ | $84.71 \pm 0.51$ | $86.76 \pm 0.52$ |
| AQ-2500 | $2138.12 \pm 6.39$ | $2050.67 \pm 5.62$ | $2121.86 \pm 14.48$ |

## D   Core code

We present the core part of AAA python code (PyTorch) below, where a_i stands for the attractor interval $\tau$ in (4), reverse_step is $\alpha$ in (5), and calibration_loss_weight is $\beta$ in (6).

```python
logits = cnn(x_curr)
logits_ori = logits.detach()
p_target = F.softmax(logits_ori / temperature, dim=1).max(1)[0]

value, index_ori = torch.topk(logits_ori, k=2, dim=1)
margin_ori = value[:, 0] - value[:, 1]
attractor = ((margin_ori / a_i).ceil() - 0.5) * a_i
l_target = attractor - reverse_step * (margin_ori - attractor)

mask1 = torch.zeros(logits.shape, device=device)
mask1[torch.arange(logits.shape[0]), index_ori[:, 0]] = 1
with torch.enable_grad():
  logits.requires_grad = True
  optimizer = torch.optim.Adam([logits], lr=optimizer_lr)

  for i in range(num_iter):
    prob = F.softmax(logits, dim=1)
    loss_c = ((prob * mask1).max(1)[0] - p_target).abs().mean()
    value, index = torch.topk(logits, k=2, dim=1)
    margin = value[:, 0] - value[:, 1]
    loss_d = (margin - l_target).abs().mean()
    loss = loss_d + loss_c * calibration_loss_weight
    optimizer.zero_grad(); loss.backward(); optimizer.step()
```

# E   Detailed experimental settings

Table 10: The used models

| Defense | Dataset | Architecture | Source | ID |
|---------|---------|--------------|--------|-----|
| None | CIFAR-10 | WideResNet-28 | RobustBench | Standard |
| PSSiLU (AT) | CIFAR-10 | WideResNet-28 | RobustBench | Dai2021Parameterizing |
| HAT (AT) | CIFAR-10 | WideResNet-34 | RobustBench | Rade2021Helper_extra |
| PNI (AT) | CIFAR-10 | ResNet-20 | Official | PNI-W (channel-wise) |
| TRS | CIFAR-10 | ResNet-20 | Official | / |
| None | ImageNet | WideResNet-50 | TorchVision | wide_resnet50_2 |
| AT | ImageNet | WideResNet-50 | RobustBench | Salman2020Do_50_2 |
| None | ImageNet | ResNeXt-101 | TorchVision | resnext101_32x8d |
| FD (AT) | ImageNet | ResNeXt-101 | Official | ResNeXt101_DenoiseAll |

**Defenses.** The detailed information of all our used models is shown in Table 10. The official repositories of PNI, TRS, and FD are `https://github.com/elliothe/CVPR_2019_PNI`, `https://github.com/AI-secure/Transferability-Reduced-Smooth-Ensemble`, and `https://github.com/facebookresearch/ImageNet-Adversarial-Training`, respectively. AAA, RND, and DENT are directly implemented on the undefended model. RND adds the random noise with variance $0.02$ to input samples as recommended in [14]. In DENT, we follow the original work to optimize the model for 6 iterations using the tent loss and Adam optimizer (lr= $0.001$). The pre-trained AT/PNI model comes from RobustBench / official repository. We train the TRS model (ensemble 3 models) using the default coeff, lambda, and scale in the official code.

**Attacks.** All the attacks are adapted from the official repositories with original hyper-parameters. SimBA and Bandit are implemented from `https://github.com/cg563/simple-blackbox-attack` and `https://github.com/MadryLab/blackbox-bandits`, respectively. SignHunter and NES are both from `https://github.com/ash-aldujaili/blackbox-adv-examples-signhunter`. Square and QueryNet are both from the implementation in `https://github.com/AllenChen1998/QueryNet`. The detailed hyper-parameters of attacks are outlined in Table 11.

Table 11: Hyper-parameters for other attacks

| Method | Hyperparameter | CIFAR-10 | ImageNet |
|--------|----------------|----------|----------|
| SimBA [4] | $d$ (dimensionality of 2D frequency space) | 32 | 32 |
| | order (order of coordinate selection) | random | random |
| | $\epsilon$ (step size per iteration) | 0.2 | 0.2 |
| SignHunter [41] | $\delta$ (finite difference probe) | 8 ([0,255]) | 0.05 ([0,1]) |
| NES [61] | $\delta$ (finite difference probe) | 2.55 | 0.1 |
| | $\eta$ (image $l_p$ learning rate) | 2 | 0.02 |
| | $q$ (# finite difference estimations / step) | 20 | 100 |
| Bandit [3] | $\delta$ (finite difference probe) | 0.1 | 0.1 |
| | $\eta$ (image $l_p$ learning rate) | 0.01 | 0.01 |
| | $\tau$ (online convex optimization learning rate) | 0.01 | 0.01 |
| | Tile size (data-dependent prior) | 50 | 50 |
| | $\zeta$ (bandit exploration) | 1.0 | 1.0 |
| Square [5] | $p$ (initial probability to change coordinate) | 0.05 | 0.05 |
| QueryNet [66] | Number of batches (NAS training) | 500 | / |
| | batch size (NAS training) | 128 | / |
| | Number of layers (NAS surrogate models) | 6, 8, 10 | / |