# OpenReview forum: "Adversarial Attack on Attackers: Post-Process to Mitigate Black-Box Score-Based Query Attacks"
_NeurIPS.cc/2022/Conference — NeurIPS 2022 Accept_

### Official Review · Reviewer_AAQp · 2022-07-09

**Rating:** 4
**Confidence:** 5
**Soundness:** 2 fair
**Presentation:** 3 good
**Contribution:** 2 fair

**Summary:**

This paper proposes an adversarial defense against black-box score-based query attack. Specifically, the AAA defense proposed in this paper can mislead SQA methods to simulate the direction of gradients by post-processing the logits output by the target model. The method is validated on cifar-10 and Imagenet datasets

**Questions:**

There seems to be some problems in the penultimate row of Table 2. The performance of 69.3 of RND defense under NES attack should be bolded, not 68.51 of AAA.

**Limitations:**

From the perspective of experimental performance, compared with other defense methods, AAA is still not significantly improved in terms of affecting SQAs’ attack effect.

**Strengths And Weaknesses:**

The strengths and weaknesses of this paper are summarized as follows:

The main strength of this paper is that it presents a relatively new approach of adversarial defense. The defense against SQA is rarely covered by previous work, and the AAA method proposed in this paper may bring inspiration to the subsequent score-based attacks and related defenses

The weaknesses of this paper are twofold. The first aspect is the motivation of the method. The AAA method proposed in this paper is in fact specially targeted for SQA. While SQA belongs to a class of query-based attacks in black-box attack methods in adversarial attacks. That is to say, the effectiveness of AAA is based on at least three constraints: the attacker cannot obtain the details of the target model, the attacker does not use a substitute model but can only query the target model, and the attacker can obtain the confidence information of the target model. This situation implies one-way transparency from the defender to the attacker, not the other way around. This setting is very unrealistic, which makes the motivation somewaht weak.

In addition, even from the perspective of exploring the robustness of deep neural networks, the method proposed in this paper to specifically deal with one class of attacks cannot improve the robustness of the model itself, or as the authors put it in the paper, worst-case robustness. However, improving the robustness of the model without considering the attack method is the direction that should be advocated in the field of adversarial defense [1]. Otherwise, this is only an arms race between adversarial attacks and defenses.

[1] Athalye, Anish, Nicholas Carlini, and David Wagner. "Obfuscated gradients give a false sense of security: Circumventing defenses to adversarial examples." International conference on machine learning. PMLR, 2018.

Secondly, because this paper defends against SQA, there is a realistic problem that other query-based attacks like Boundary Attack require less information than SQA (no confidence information, just hard labels). Can the method proposed in this paper effectively defense against such attacks? If so, AAA must changes the hard label, will this affect the availability of the target model? If not, can the attacker directly use decision-based attacks to bypass this defense?

---

> ### Author Response · Authors · 2022-08-02
> **Response to Reviewer AAQp**
>
> Thanks for appreciating the novelty of our work. Before our response, we would like to thank reviewer AAQp for highlighting the defense against all unseen attacks. Indeed, making DNNs truly robust in the worst case is significant for the community to guarantee the non-existence of adversarial examples and make DNNs explainable and interpretable from a theoretical aspect. From an application aspect, similarly, studying defenses feasible only in real-world scenarios also makes a difference to secure DNNs, seeing a recent defense especially against SQAs [a]. We address other concerns as below:
>
>
> **R4.1 Three unrealistic constraints of SQAs make especially mitigating SQAs a weak motivation**
>
> Let us consider that in real-world applications, attackers and defenders are in a double-blind relationship. Thus, attackers naturally have real-world constraints as widely highlighted [b,c,d].
>
> (1) Target model is unobtainable because revealing model details reduces the commercial competence of model owners [e]. (2) Substitute model is also unobtainable [b,c,d] because it needs the target model’s training data, and leaking user data violates the company's privacy commitments. Attackers can indeed train a substitute model by querying the target model, but it would require tens of thousands of queries to the target model [f]. (3) Model owners would reveal its prediction confidence, seeing Google Cloud Vision API, Baidu EasyDL, etc., because model uncertainty information is crucial for users to make downstream decisions [g].
>
> We understand that objective real-world constraints on attackers are not commonly seen by defenders because defending against SQAs is rarely explored as you mention. But that is exactly why our new take on such a field is valuable as also acknowledged by Reviewer 78Et.
>
>
> **R4.2 AAA defense under decision-based attacks (DQAs)**
>
> Yes, this is one limitation of our method (discussed on **Line 334 in the original submission**), i.e., AAA cannot mitigate DQAs because it does not change the model’s decision. However, that does not hurdle the significance of defending against the much more threatening SQAs.
>
> As you mention, attackers could either use SQAs or decision-based attacks (DQAs) in real-world scenarios. With SQAs, attackers could greatly decrease the model’s performance within dozens of queries [b,c,d]. Thus, defending especially against such threatening attacks is well motivated as acknowledged by Reviewer uWUQ.
>
> With DQAs, however, model accuracy could not be perceptibly influenced within thousands of queries [h,i,j] due to the lack of model scores, as we demonstrate in **Appendix A in the original submission**. In this regard, especially defending against SQAs is meaningful despite AAA's failure in mitigating DQAs.
>
>
> **R4.3 AAA does not significantly hurdle SQAs**
>
> We respectfully disagree with Reviewer AAQp on the insufficiency of AAA's performance. According to Fig 3, AAA prevents the adversarial accuracy from dropping quite effectively, doubling the performance of AT and tripling that for RND. Table 2 also supports the non-trivial protection ability of AAA against 8 defenses. Although in rare (2 out of 30 thanks to your careful reading) cases AAA does not top the performance, AAA significantly outperforms baselines in accuracy (no decrease), calibration (improvement), and defense costs (no cost on training and negligible cost on testing), seeing Table 1 and Table 2. Such effective defenses against SQAs have not been proposed to the best of our knowledge. The good performance of AAA is also acknowledged by all other reviewers, e.g., "AAA shows strong performance on a variety of settings" by **Reviewer 78Et**.
>
> **References**
>
> [a] Z. Qin, Y. Fan, H. Zha, and B. Wu, “Random noise defense against query-based black-box attacks,” NeurIPS 2021.
>
> [b] C. Guo, J. Gardner, Y. You, A. G. Wilson, and K. Weinberger, “Simple black-box adversarial attacks,” ICML 2019
>
> [c] M. Andriushchenko, F. Croce, N. Flammarion, and M. Hein, “Square attack: A query-efficient black-box adversarial attack via random search,” iECCV 2020
>
> [d] A. Al-Dujaili and U.-M. O’Reilly, “Sign bits are all you need for black-box attacks,” ICLR 2019.
>
> [e] Li Y, Zhu L, Jia X, et al. Defending against model stealing via verifying embedded external features, AAAI 2022.
>
> [f] Zhou, Mingyi, et al. "Dast: Data-free substitute training for adversarial attacks." CVPR 2020.
>
> [g] C. Guo, G. Pleiss, Y. Sun, and K. Q. Weinberger, “On calibration of modern neural networks,” ICML 2017
>
> [h] Brendel W, Rauber J, Bethge M. Decision-Based Adversarial Attacks: Reliable Attacks Against Black-Box Machine Learning Models, ICLR 2018.
>
> [i] D. C. R. Viet Quoc Vo, Ehsan Abbasnejad, “Ramboattack: A robust query efficient deep neural network decision exploit,” NDSS, 2022.
>
> [j] M. Cheng, S. Singh, P. H. Chen, P.-Y. Chen, S. Liu, and C.-J. Hsieh, “Sign-opt: A query-efficient hard-label adversarial attack,” ICLR, 2019.

---

### Official Review · Reviewer_uWUQ · 2022-07-09

**Rating:** 7
**Confidence:** 4
**Soundness:** 4 excellent
**Presentation:** 3 good
**Contribution:** 4 excellent

**Summary:**

This paper proposes a post-processing defense against score-based black-box evasion attacks. This defense modifies the original logits so that the trend of the margin loss is reversed in each small interval of the attack, but the overall trend remains unchanged to preserve accurate prediction confidence. As a result, score-based attacks using the margin loss will optimize towards the opposite (hence non-adversarial) direction. Meanwhile, this defense does not hurt the model’s benign performance, and the calibration performance is preserved by solving a joint optimization problem. Experiments show that the proposed defense outperforms 8 previous defenses under 6 score-based attacks on CIFAR-10 and ImageNet.

**Questions:**

My current score is mainly based on the weaknesses outlined in the Quality section. I am willing to raise my score if the following concerns are adequately clarified or justified.
* [Quality-Weakness-1] Please clarify if the knowledge of the attack’s period $t$ at L171 is practical. Specifically, please explain how the proposed defense can be implemented *independently of the attack’s code* with only the following inputs: (1) query image, (2) model, and (3) the defense’s hyper-parameters. If you need any other inputs, please clarify relevant assumptions and their practicability.
* [Quality-Weakness-2] Please discuss if the defense would still work if the attackers revert their update logic in Eq. (3), without significantly modifying the current defense.

**Limitations:**

The primary limitation is summarized in the Significance section. While I appreciate the effort of motivating post-processing defenses against evasion attacks, I strongly recommend the authors carefully address these limitations in future versions of this paper.

**Strengths And Weaknesses:**

### Originality

**Strengths**
* This paper studies post-processing defenses, which are less explored in defending evasion attacks.
* The proposed defense avoids hurting the model’s benign performance and computational overheads.

**Weaknesses**
* Not much in this perspective.

### Quality

**Strengths**
* Good experiments. I appreciate the evaluation of 8 defenses, 6 attacks, and 2 datasets.
* Good ablation study. Most hyper-parameters are supported by the ablation study.

**Weaknesses**
* **Impractical threat model.** While the attack’s threat model is well defined, the proposed defense implicitly assumes complete control of the attacker’s optimization process. **Specifically, the defense assumes unrealistic knowledge of the attack’s running iteration, i.e., the period $t$ defined at L171**. Note that all compared defenses in this paper did not have such an assumption. This knowledge is only realizable if given the following two assumptions. First, the attacker discloses the attack’s current number of iterations for each query. Second, the defender can extract the sequence of queries coming from the designated attacker (across all queries from all legitimate users and potentially other attackers). These two assumptions, however, are largely unrealistic or challenging. Since such knowledge is the base assumption of this defense at Eq. (4), I am not sure if the current defense would work if the assumption did not hold. **In particular, the defense’s code in Appendix D seems to work inside the attack, which is impossible in practice.**

* **Lack of adaptive evaluation.** This paper does not discuss how the attacker could potentially modify their attacking procedure to evade the proposed defense, although some adaptive attack papers like [46] are cited. Currently, the proposed defense assumes a static (i.e., non-adaptive) attack that is unaware of the defense. Following the paper’s own motivation, is it possible that the attackers also revert their updating logic in Eq. (3) to evade the proposed defense? For example, now that the returned score exhibits a reverted trend, **would this defense still work if the attackers update their adversarial examples only when observing an increased margin loss?**

### Clarity

**Strengths**
* Good motivation for post-processing defenses.
* The proposed defense is easy to follow.

**Weaknesses**
* The notation in Eq. (1) is slightly confusing. At L117, the x is defined as a clean sample, but it is later used as a placeholder in Eq. (1) and (2).
* The unsupervised margin loss in Eq. (2) can be simplified. Specifically, if the defender could use one query to obtain the ground-truth label (from the black-box model’s perspective), Eq. (2) can reduce to Eq. (1) and therefore simplify the notations.
* At L123, it is suggested to add some (brief) background of how these attacks sample their queries $x_q$. This would be greatly helpful for readers to understand the defense. Currently, it seems that all attacks are compressed into one high-level idea, which might reduce the confidence in understanding how the defense would work if the attack changes adaptively.
* At L171 and L193, the notation $z_0$ is not defined until algorithm 1.

### Significance

I appreciate the effort of motivating post-processing defenses against evasion attacks, but the proposed defense considers a somewhat unrealistic threat model, which is different from the defenses compared in this paper. Moreover, this paper assumes a static non-adaptive attacker, which might be easily broken by adaptive attacks based on my assessment.

---

> ### Author Response · Authors · 2022-08-02
> **Response to Reviewer uWUQ**
>
> Thanks for highly appreciating the originality and experiments of our work. We address your concerns as below:
>
> **R3.1 AAA seems to unrealistically require knowledge of attack iterations, seeing the t in Eq (4)**
>
> It is a misunderstanding. AAA does NOT need the attack running iterations. The t in Eq (4) stands for the period of loss intervals, a constant hyper-parameter of AAA that defenders set as explained on **Line 169-171 (original submission)**, and it is independent and irrelevant to the attack process.
>
> After receiving the query image, AAA uses the model to conduct normal inference and calculates the original loss value $l_0$ by Eq (2). Then AAA uses the pre-set hyper-parameter t to decide the target loss value $l_t$ by Eq (4) and optimizeS as Eq (6). Since t simply divides the original loss values into intervals [0, t], [t, 2t], [2t,3t], … and it is obvious that defenders do not need the attack running iterations to calculate the loss, AAA requires no impractical knowledge of attackers. We hope that making the notation t more clear in the **revised version (Line 172)** avoids future confusion.
>
> Given the analysis above, defenders do not peek at the attacker's inner state. And vice versa, attackers do not know about the defender's strategy, forming the realistic double-blind real-world threat model.
>
>
> **R3.2 AAA on defending adaptive attacks, e.g., search following an increased margin loss**
>
> Designing adaptive attacks, though possible, is costly and easy to bypass in real-world scenarios. Because here, according to **R3.1**, attackers and defenders are in a double-blind relationship, i.e., attackers do not know the model, including the defense strategy.
>
> Thus, the discovery process of defense strategies for developing adaptive attacks would require additional queries and creative deductions. For example, attackers have to first query lots of times before observing the rare exceptional loss change. Then deduce the whole unknown reversing loss strategy and its hyper-parameters with great manual efforts. Only after such endeavor could SQA attackers decide to go opposite as you creatively mention.
>
> Moreover, attackers are also foolable if they go in case of increased loss or dramatically decreased loss (to jump out of an interval). Because we could simply smooth the transition between intervals, e.g., by using a sine function to design the target loss following the same idea of confounding attackers. In this way, neither direction of search is likely to figure out the defense strategy, seeing the results below.
>
> Table A: AAA performance under adaptive attacks (100 queries)
> | Defense methods  | None  | AAA   | AAA-sine |
> | :------------ | :-----------: | :-----------: | :-----------:  |
> | Square | 39.38 |  81.36 | 79.78  |
> | opposite Square | 94.78 | 57.31 | 75.23 |
>
> The opposite search weakens AAA from 81.36\% to 57.31\%, but motivated by you, we could use a sine-like function to let the loss periodically ascend and descend along the original attack direction. In this way, the defense performance for the regular attack is kept as 79.78\% while the performance under the opposite search is improved to 75.23\%.
>
> Our main contribution lies in the philosophy to confound real-world attackers by a misleading but slightly-modified loss trend, rather than the specific strategy. And therefore, there are lots of other designs, and defenders could flexibly switch between them to mislead the attacker's guessing. Thus, AAA is effective even if attackers try to develop adaptive strategies. Relevant discussions have been added to the **revised submission (Line 308-328)** thanks to your insightful comments.
>
>
>
> **R3.3 Clarity of equations, notations, and SQA backgrounds**
>
> Thanks a lot for your careful reading. In the revised version, we have replaced “clean sample x” as “sample x” for Eq (1) **(Line 117)**, simplified Eq (2) by setting ground truth label in view of defenders **(Line 120)**, explained z0 the first time it appears **(Line 171)**, and provided introductions of several specific SQAs **(Line 220-224)**. We are very grateful that you are willing to conduct such insightful and detailed discussions and consider raising the score.

---

> > ### Comment · Reviewer_uWUQ · 2022-08-05
> > **Thank you for your detailed response!**
> >
> > I appreciate the detailed clarifications, and all my concerns are clarified.
> >
> > **R3.1 Misunderstanding.**
> >
> > Thank you for the clarification, it makes more sense now. I believe the misunderstanding came from misreading $l_0$ as a constant and $t$ as the step variable in Eq. 4 (since $t$ was redefined and used as a subscript in Eq. 5 and all these subscripts in Eq. 5 quite looked like losses at different steps).
> >
> > I would recommend revising the notations to avoid this confusion. For example, use $\tau$ for the period, spell out $l_\mathrm{a}$ and $l_\mathrm{t}$ as $l_\mathrm{attract}$ and $l_\mathrm{target}$ (or something else). It might also be good to emphasize which symbols are variables (wrt the input), it is very easy (at least for me) to misinterpret $l_0$ and $z_0$ as global constants, rather than variables that change corresponding to the query sequence.
> >
> > For another statement in my initial review, *"extract the sequence of queries coming from the designated attacker"*, it is suggested to clarify that the proposed defense does not need to have different actions for benign and adversarial queries, as it won't affect benign queries much.
> >
> > **R3.2 Adaptive attacks.**
> >
> > The proposed sine variant is insightful and convincing. It should withstand a reasonable level of adaptive attacks to some extent. Given its higher robustness, I would recommend highlighting this variant earlier in the paper. I personally think that this variant is more mature than the original defense (and should have been the original proposal).
> >
> > **Summary**
> >
> > Given the above clarifications, I am switching my score from 2 to 7.
> >
> > The strengths were in my original review. My initial negative score was mainly due to a major misunderstanding, as explained above. I am also not too worried about decision-based and transfer-based attacks (raised by other reviewers), as they are not claimed as contributions, and I am satisfied with the current contribution.

---

> > > ### Author Response · Authors · 2022-08-06
> > > **Thank you for reconsidering our work!**
> > >
> > > We would like to express our sincerest thanks to Reviewer uWUQ for the efforts to re-evaluate and (again) improve our work. Besides, we are grateful to see Reviewer uWUQ understand that our contributions lie beyond defending against decision-based and transfer-based attacks.
> > >
> > > **R3.1 Revision of notations**
> > >
> > > It would be undoubtedly clearer to distinguish the loss attractor $t$ from the “target” subscript $l_t$. Following your constructive suggestions, we have modified the notations with specifications on variables to avoid future confusion.
> > >
> > > Concretely, we denote the period as $\tau$, and spell out $l_a, l_t, l_0, z_o, p_t$ as $l_\mathrm{atr}, l_\mathrm{trg}, l_\mathrm{org}, z_\mathrm{org}, p_\mathrm{trg}$, respectively in the revised version. Besides, we also note in Alg. 1 that only $T, \tau, \alpha, \beta$, and the number of optimization iterations $\kappa$ are constant hyper-parameters.
> > >
> > > We hope that readers, especially those interested in our specific method, could easily grasp our notations.
> > >
> > >
> > > **R3.2 Highlighting the sine design of AAA earlier**
> > >
> > > Thanks for your careful thought. Indeed, our contribution lies in the philosophy to confound attackers by slight output perturbations, which requires a periodic design. Thus, the specific strategy to control the in-interval loss trend indeed has many choices, e.g., the linear and the sin curves.
> > >
> > > Inspired by the discussions with you, we plan to put the emphasis on the periodic design and include different specific functions, especially AAA-sine. We think the logic of your suggestion is more nature and the modification is not too much. We will try our best to prepare a rigorous revised version.

---

### Official Review · Reviewer_78Et · 2022-07-10

**Rating:** 7
**Confidence:** 4
**Soundness:** 3 good
**Presentation:** 4 excellent
**Contribution:** 4 excellent

**Summary:**

This paper proposes a defense called Adversarial Attack on Attackers (AAA) that is designed specifically towards mitigating score-based query attacks (SQAs). AAA is a post-processing attack that attempts to modify the logits loss curve to locally point in the incorrect attack direction in a periodic fashion, which steers SQAs away from a true adversarial attack. As such, AAA takes advantage of typical SQA behavior of sampling nearby loss value changes to find a local direction to optimize in and steers them in the wrong way. AAA could then be added on to other models as a way to deter and prevent these attacks in more real-world scenarios where such black-box access is present. The authors evaluate AAA over several baseline defenses on CIFAR-10 and ImageNet on various SQAs, finding AAA to be effective at improving robustness.

**Questions:**

This paper provides an interesting new take on adversarial robustness. I like the direction of targeting realistic attacks such as SQAs at a higher level than simply robustifying the model to all possible attack inputs. Indeed, such a post-processing approach such as AAA to deter attacks that that level is appealing and I believe that further research with this philosophy would be interesting. The experiments presented provide a useful understanding of the characteristics of AAA and show strong performance on a variety of settings.

I have two main concerns. The primary one is whether or not SQAs could possibly be adapted to attack AAA. In Section 5.5 “Hyper-parameters of AAA” and Figure 4, it is clear that the choice of the attractor interval t can greatly influence the adversarial accuracy as the attack success rate would seemingly come down to the attack’s ability to jump out of an interval. While this is an interesting study to see how the choice of t impacts the performance of a static SQA, could SQAs be modified to look at a wider interval?

For example, perhaps an SQA could be modified to maintain two search branches, one that goes in the default direction implied by the loss changes and another that continually goes in the opposite for a limited time, and then if the one that goes in the opposite direction eventually drops by jumping out of the interval, an attacker could determine that this was actually the correct direction to go. Since the underlying model is not inherently more robust, a process that finds a jump may still be able to eventually break the model, albeit with more queries.

Secondarily, how much does it cost to run AAA? It would appear to be a fairly lightweight process, but do you know how much slower this would make the model?

Given this, I still like the general direction of the paper and if my concerns are addressed I will consider raising my score.

-----------------------------------------------

In the rebuttal, the authors provided results on the proposed bidirectional adaptive attack as well as a new sine function to use, as well as runtime results, alleviating these concerns.

**Limitations:**

The authors have adequately addressed limitations and impact. The authors discuss AAA in the context of real-world systems such as autonomous driving where AAA could be added with pre-trained models to increase the reliability of such systems. The authors also acknowledge the limitation that AAA is designed to target SQAs specifically, which is a useful defense approach, but does mean that worst-case robustness under white-box settings is not improved.

**Strengths And Weaknesses:**

Strengths
- Interesting new defense approach that targets a realistic deployment of ML models by attacking the query process
- Seems like a lightweight defense with a low cost
- Preserves natural accuracy

Weaknesses
- Unclear if SQAs could possibly be adapted to re-attack AAA
- Unknown what the exact runtime cost is
- Does not increase underlying model robustness

---

> ### Author Response · Authors · 2022-08-02
> **Response to Reviewer 78Et**
>
> Thanks for appreciating our new direction in defending against realistic SQAs. We address your concerns as below:
>
>
> **R2.1 AAA on defending against adaptive attacks, e.g., bidirectional search to jump out of an interval**
>
> Designing adaptive attacks, though possible, is costly and easy to bypass in real-world scenarios. Because here, attackers and defenders are in a double-blind relationship, i.e., attackers do not know the model, including the defense strategy.
>
> Thus, the discovery process of defense strategies for developing adaptive attacks would require additional queries and creative deductions. For example, the interesting bidirectional attack strategy you propose uses extra queries to probe the exceptional loss change, which is rare if attackers act in a common way. After that, attackers also have to devote considerable manual efforts to figure out what defenders actually do. In this regard, AAA imposes a great hurdle to adaptive attackers.
>
>
> Moreover, attackers are also foolable if they base their action on the signal of jumping out of an interval. Because such a signal may not be sensible if we simply smooth the transition between intervals, e.g., by using a sine function to design the target loss following the same idea of confounding attackers. In this way, neither direction of search is likely to figure out the defense strategy, seeing the results below.
>
> Table A: AAA performance under adaptive attacks (100 queries)
> | Defense methods  | None  | AAA   | AAA-sine |
> | :------------ | :-----------: | :-----------: | :-----------:  |
> | Square | 39.38 |  81.36 | 79.78  |
> | bidirectional Square | 57.09 | 62.91 | 75.36 |
>
> The bidirectional search weakens AAA from 81.36\% to 62.91\%, but motivated by you, we could simply use a sine-like function to let the loss ascend and descend along the original attack direction. In this way, the defense performance for the regular attack is kept at 79.78\% while the performance under the bidirectional search is improved to 75.36\%.
>
> Our main contribution lies in the philosophy to confound real-world attackers by a misleading but slightly-modified loss trend, rather than the specific strategy. And therefore, there are lots of other designs, and defenders could flexibly switch between them to mislead the attacker's guessing. Thus, AAA is effective even if attackers try to develop adaptive strategies. Relevant discussions have been added to the **revised submission (Line 308-328)** thanks to your insightful comments.
>
>
> **R2.2 AAA runtime on device**
>
> Thanks for bringing in this concern. We report AAA’s actual runtime by looking at the balance between the amount of computation (number of optimization iterations) and defense performance in CIFAR-10 experiments.
>
> Table B: Influence of the optimization times in AAA (100-query Square attack on CIFAR-10)
> | No. iter | 0 | 20 | 40 | 60 | 80 | 100 |
> | :------------ | :-----------: | :-----------: | :-----------:  | :-----------: | :-----------: | :-----------:  |
> | ECE | 3.52 | 2.87 | 2.81 | 2.66 | 2.53 | 2.53 |
> | Adv-Acc | 39.38 | 79.29	| 80.92| 81.37	| 81.28	| 81.36
> | inference time per sample (ms) |1.016 | 1.034 | 1.088 | 1.099 | 1.143	| 1.163
>
> As we can see, optimizing low-dimensional logits is not costly, which thus has already become a common practice in model calibration [32, 57]. It only consumes 1.5s to optimize 10000 logits for 100 iterations (AAA’s default setting) in an NVIDIA Geforce RTX 2080Ti. And good defense and calibration results are also obtainable by 60-80 iterations, which costs even less time. Since optimizing logits is independent of model size, model owners could determine AAA runtime precisely.
>
> Relevant contents have been put in the **revised submission on Line 298-303**. We are very grateful that you engage in such insightful discussions and consider raising the score.

---

> > ### Comment · Reviewer_78Et · 2022-08-05
> > **Response to Authors**
> >
> > Thank you for your detailed rebuttal. In particular, the introduction of the sine function is very interesting and I am glad to see results on the bidirectional adaptive attack idea, both with the larger drop in performance initially and with the quite modest drop in performance after adding the sine function. The addition of these results make the paper stronger and addresses my adaptive attack concerns. The runtime results are also very encouraging.
> >
> > I agree with the updated opinion of reviewer uWUQ and think this is a good submission with these changes. I have thus accordingly updated my score to 7 as well.

---

> > > ### Author Response · Authors · 2022-08-06
> > > **Thank you for re-evaluating our work!**
> > >
> > > We are very grateful for your efforts in re-considering our sine design and adaptive attacks. We would keep improving our manuscript to involve your insights.

---

### Official Review · Reviewer_cULW · 2022-07-13

**Rating:** 7
**Confidence:** 3
**Soundness:** 3 good
**Presentation:** 3 good
**Contribution:** 3 good

**Summary:**

The present work introduces a novel defense to confound the score-based query attacks (SQAs). Its main novelty includes three parts: (1) an effective and user-friendly post-processing module; (2) a novel adversarial attack on attackers (AAA) defense by slightly deviating the output scores; (3) AAA beats other prior defenses in terms of the effectiveness and robustness. Due to the impractical cost of prior defenses, AAA designs the post-processing method to improve efficiency and exerts a significant impact on accuracy.

**Questions:**

1. Interesting, the clean model performance with AAA seems a little higher than original clean model, for example, 94.84 vs 94.78 in CIFAR10. Can you provide more analysis about this? Similar results can also be found in baseline DENT.

2. In Equation (4), does it missing abs for ceil(·)? when l_0=0, ceil(l_0/t=0)-1/2 = -1/2? and the result is -2?

3. The idea of attacking on attacker is straightforward, what is the motivation in designing the specific operated shown in Equation (4) and (5). For example, why push l_t even lower than l_a by a margin of (l_0-l_a), and what is the detailed motivation in designing equation (4). I am just curious if there are any other operations following the same idea achieves better defense results.

4. Have you studied the proposed method without optimization in Equation (6)?

5. The adjusted loss is closely related with confidence calibration. Can you show some defense results, in comparison with some confidence calibration methods?

**Limitations:**

The discussion about optimizing Equation 6 seems missing. What is the number of optimization steps? The time consumed? etc. What are the results without optimization Equation (6).

**Strengths And Weaknesses:**

Strengths:
+ The idea is sound and interesting. Instead of enhancing the robustness of deep learning models, this proposed method mislead and confuse the attackers to protect the DNN.
+ Meanwhile, the method costs a little to improve efficiency, which makes defense more practical in real-world settings.
+ the general design of the experiments is meaningful in helping to understand the role of AAA in outperforming the considered defense approaches.

Weaknesses:
- Some operations for example, the determination of specific operation, like Equation (4) is short of discussion. The detail is as follows.

---

> ### Author Response · Authors · 2022-08-02
> **Response to Reviewer cULW**
>
> Thanks for your insightful comments and appreciation of the novelty. We address your concerns as below:
>
>
> **R1.1 Motivation of Eq (4)(5)**
>
> Our design is motivated by two goals of real-world defenses: fooling attackers and serving users. The former requires reversing the loss trend along the attack direction while the latter demands slight changes in the loss value. More importantly, we found these two (seemingly contradictory) goals can be reconciled if we instantiate them locally, i.e., first divide loss values into intervals by Eq (4), and then reverse the loss trend in each interval by Eq (5). Thus, Eq (4) is necessary to ensure slight output perturbations.
>
> Our main contribution lies in the philosophy to confound real-world attackers by a misleading but slightly-modified loss trend, rather than the specific strategy. And therefore, there are indeed lots of designs besides Eq (5) to fool SQAs in each interval as you wisely guess. For example, we could map original loss to target loss in a sine way. Since attackers and defenders are double-blind without knowledge about each other in real-world scenarios, either design could effectively fool SQAs (even adaptive attacks) well as discussed on **R2.1, R3.2, and the revised version.**
>
> As for Eq (5), it maps a large original loss $l_0>l_a$ to a small target loss $l_t<l_a$, and vice versa. In this way, when the original loss decreases from $l_a + t/2$ to $l_a - t/2$, AAA outputs loss that increases from $l_a - \alpha * t/2$ to $l_a + \alpha * t/2$.
>
>
> **R1.2 Eq (4) misses abs when $l_0=0$**
>
> Thanks for your insightful thinking. Although $l_0=0$ is only a corner case since $l_0 \geq 0$, it would be more rigorous to add an abs to Eq (4), i.e., $l _ { a } = ( \operatorname { floor } ( l _ { 0 } / t ) + 1 / 2 ) \times t$ in the revised version.
>
>
> **R1.3 Discussions of Eq (6) on formulation, optimization, and runtime**
>
> Eq (6) is a straightforward design to fulfill the two goals that we discussed in **R1.1**. First, we perturb the logits to have a margin loss $L_u(z)$ close to the target value $l_t$, forming the reversed loss curve to fool SQAs. Accordingly, the first term in Eq (6) is formulated to minimize the distance between $L_u(z)$ and $l_t$. Second, we want the modified logits to output confidence ($\sigma (z)$, the maximum probability after softmax) close to the calibrated one $p_t$ so that users get accurate confidence scores. In this regard, the second term in Eq (6) is designed to minimize the distance between $\sigma (z)$ and $p_t$. The $\beta$ balances the optimization between the above two goals.
>
> Despite its simplicity, Eq (6) has to be solved by optimization because the exponential operation in softmax makes Eq (6) a transcendental equation without closed-form solutions. Luckily, optimizing low-dimensional logits is not costly, which has already become a common practice in model calibration [32, 57]. Please see the results below.
>
> Table A: Influence of the optimization times in AAA (100-query Square attack on CIFAR-10)
> | No. iter | 0 | 20 | 40 | 60 | 80 | 100 |
> | :------------ | :-----------: | :-----------: | :-----------:  | :-----------: | :-----------: | :-----------:  |
> | ECE | 3.52 | 2.87 | 2.81 | 2.66 | 2.53 | 2.53 |
> | Adv-Acc | 39.38 | 79.29	| 80.92| 81.37	| 81.28	| 81.36
> | inference time per sample (ms) |1.016 | 1.034 | 1.088 | 1.099 | 1.143	| 1.163
>
> If we choose the default optimization iterations (100 as shown on **Line 213 in the original submission**), it only consumes 1.5s to optimize 10000 logits in an NVIDIA Geforce RTX 2080Ti. And good defense and calibration results could also be obtained by 60-80 iterations, which costs even less time. Since the time for optimizing logits is independent of model size, model owners could determine its runtime very precisely. A study on AAA's runtime is on the **new version**.
>
> **R1.4 Explanations on accuracy increase by AAA**
>
> Thanks for pointing out this phenomenon. We re-run experiments multiple times with different seeds (for optimization) and hyper-parameters, and find that AAA’s accuracy cannot stably outperforms the original baselines (e.g., usually oscillating very slightly above or below). Therefore, we conclude that such a small difference comes from randomness and may not worth further discussion.
>
> **R1.5 Defense results by calibration methods**
>
> Thanks for bringing up this comparison. Note that calibration is supposed to map $p_1 > p_2$ to $p'_1 > p'_2$ without reversing the loss trend along the attack direction. Thus, attackers can still steal the gradient and attack, seeing our additional test of a set of standard calibration baselines below.
>
> Table B: SQA Defense performance by calibration methods (30-query Square attack on CIFAR-10)
> | Calibration methods  | None  | temperature scaling [32]  | histogram binning [57] |
> | :------------ | :-----------: | :-----------: | :-----------:  |
> | ECE | 3.52 |  2.02 | 0.78 |
> | Adv-Acc | 68.85 | 66.81 | 68.26 |

---

> > ### Author Response · Authors · 2022-08-09
> > **Modification of manuscript following your comments**
> >
> > Dear Reviewer cULW,
> >
> > Thanks for your detailed suggestions again. We understand that the discussion period is short. And it would be time-consuming for you to inspect the response in detail. Thus, we summarize our modifications of the paper in your advice again, hoping to receive your feedback.
> >
> > **R1.1 Motivation of Eq (4)(5)**
> >
> > We add a significant number of paragraphs (Line 162-183) to clarify the formulation and necessity of Eq (4)(5). Motivated by you, it is interesting to see that it is necessary to divide the loss values into intervals by Eq (4), but there are various choices to fool SQAs in an interval as Eq (5) shows.
> >
> > **R1.2 Eq (4) misses abs when $l_0=0$**
> >
> > We have made the equation more rigorous in Line 169 thanks to your suggestions.
> >
> > **R1.3 Discussions of Eq (6) on formulation, optimization, and runtime**
> >
> > Line 189-195 and Line 302-307 are additional discussions on Eq (6) besides our initial response. We sincerely thank you for making the paper more solid significantly.
> >
> > **R1.4 Explanations on accuracy increase by AAA**
> >
> > Motivated by your, the interesting discovery that it comes from randomness has been mentioned in Line 259.
> >
> > We sincerely hope you could check your great help in our work and re-evaluate them.
> >
> > Best wishes,
> >
> > Anonymous author(s) of Paper2763

---

### Author Response · Authors · 2022-08-02
**General Response**

Dear Program Chairs, Area Chairs, and Reviewers,

First of all, we would like to thank you for your time, constructive critiques, and valuable suggestions, which greatly help us improve the work. We are also grateful that reviewers unanimously regard our work as novel and interesting. The concerns are mainly focused on attackers’ and defenders' knowledge of each other. Below we would like to first respond to issues concerning our threat model in general.

In real-world scenarios we focus on, attackers do not access the defender's model gradients, training data, and defense strategies. Therefore, the constraints on attackers (inaccessibility to model details and substitute models) mentioned by Reviewer AAQP are realistic. Moreover, without knowing the defender's strategy, attackers cannot easily design adaptive attacks as Reviewer 78Et and uWUQ bring in. To guess the defender's strategy, additional queries and creative deduction are required. And such guessing could be greatly complicated in an easy way for defenders (R1.1, R2.1, R3.2). In this regard, the great hurdle AAA imposes on even adaptive attackers verifies the significance of our work.

Defenders also do not know whether a query is malicious, what the attack method is, and when the attack has proceeded. Thus, accurate decisions are preferable, though preserving decisions makes AAA not able to mitigate decision-based attacks as Reviewer AAQP guesses, it does not reduce the significance of our defense against the more threatening score-based attacks (R4.2). Additionally, AAA does not require knowledge of the attack iterations as Reviewer uWUQ misunderstood (R3.1).

More in-depth analysis concerning the threat model (R2.1, R3.2, R4.1), AAA time-efficiency (R1.3, R2.2), and other issues have been added in the revised submission with red markers. We sincerely look forward to further discussions with the reviewers.

Best wishes,

Anonymous author(s) of Paper2763

---

### Author Response · Authors · 2022-08-09
**Summary of paper modifications**

Dear Program Chairs, Area Chairs, and Reviewers,

Thanks for the constructive comments and helpful discussions, we have carefully modified the manuscript according to the reviewers’ suggestions.

- Descriptions on AAA-sine for adaptive attacks **(Line 53-56, 62-64, 162-183, 216-219, 312-332)**
- Discussions on the motivation of Eq (4)(5)(6) **(Line 162-195)**
- Study of time consumption **(Line 302-307)**
- Clarification of notations **(changed most subscripts, Line 203)**
- Moving additional defense results and hyper-parameter study to Appendix **(Line 650-676)**

Since our main contribution lies in the philosophy to confound real-world attackers by a misleading but slightly-modified loss trend, rather than the specific strategy, the new version does not contain substantial changes to our claim but provides more analysis, discussions, and clarifications thanks to the reviewer’s insightful feedback.

Currently, we have addressed all concerns (e.g., adaptive attacks, time consumption) from Reviewer uWUQ and Reviewer 78Et thanks to their active feedback and careful re-evaluation. We are enthusiastically eager to discuss with Reviewer cULW and Reviewer AAQp since we are not allowed to respond to your valuable opinions tomorrow.

Best wishes,

Anonymous author(s) of Paper2763

---

### Meta-Review · Area_Chair_UkKX · 2022-08-27

**Recommendation:** Accept
**Confidence:** Certain

**Metareview:**

This paper proposes a defense against score-based black-box attacks by post-processing the output probabilities to misguide the attacker. The method enjoys several advantages such as not reducing test-time accuracy or increasing the train-/test-time cost, improving calibration for the model, and superior performance under black-box attack compared to prior work. The authors also included additional experiments during the discussion phase that show effectiveness against adaptive attacks.

One weakness is that the method does not improve robustness of the underlying model and hence is still susceptible to surrogate model and/or hard-label attacks. However, most reviewers consider this weakness as minor and that the paper’s contribution is significant enough for publication. AC therefore recommends acceptance for publication at NeurIPS.


**Award:**

No

---

### Decision · Program_Chairs · 2022-09-14

Accept